# Biodiversity and Variations of Arbuscular Mycorrhizal Fungi Associated with Roots along Elevations in Mt. Taibai of China

Mengge Zhang [1,2,3,†], Mei Yang [1,2,3,†], Zhaoyong Shi [1,2,3,*], Jiakai Gao [1,2,3] and Xugang Wang [1,2,3]

1   College of Agriculture, Henan University of Science and Technology, Luoyang 471023, China
2   Luoyang Key Laboratory of Symbiotic Microorganism and Green Development, Luoyang 471023, China
3   Henan Engineering Research Center of Human Settlements, Luoyang 471023, China
*   Correspondence: 9903105@haust.edu.cn
†   These authors contributed equally to this work.

**Abstract:** (1) Background: environmental gradient strongly affects microbial biodiversity, but which factors drive the diversity of arbuscular mycorrhizal fungi (AMF) associated with roots at relatively large spatial scales requires further research; (2) Methods: an experiment on large spatial scales of Mt. Taibai was conducted to explore the biodiversity and drivers of AMF-associated with roots using high-throughput sequencing; (3) Results: a total of 287 operational taxonomic units (OTUs) belong to 62 species representing 4 identified and 1 unclassified order were identified along different altitudinal gradients. With increasing altitude, AMF colonization could be simulated by a quadratic function trend, and altitude has a significant impact on colonization. AMF alpha diversity, including the Sobs and Shannon indexes, tended to be quadratic function trends with increasing altitude. The highest diversity indices occurred at mid-altitudes, and altitude had a significant effect on them. AMF communities have different affinities with soil and root nutrient, and *Glomus* is most affected by soil and root nutrient factors through the analysis of the heatmap. *Glomus* are the most dominant, with an occurrence frequency of 91.67% and a relative abundance of 61.29% and 53.58% at the level of species and OTU, respectively. Furthermore, AMF diversity were mostly associated with soil and root nutrients; (4) Conclusions: in general, AMF molecular diversity is abundant in Mt. Taibai, and altitude and nutrient properties of soil and root are the main influencing factors on AMF diversity and distribution.

**Keywords:** arbuscular mycorrhizal fungi (AMF); diversity; community; altitude; mountain





## 1. Introduction

Mountain ecosystems are rich in species diversity, and the climatic gradients are obvious within a relatively short distance, so it provides more possibilities for the research of biodiversity [1]. The diversity and community distribution of plants and animals are most frequently investigated in mountain ecosystems because of the environmental gradients and slant characteristics on a small spatial scale [2]. Peters et al. (2016) reported that the species richness of nearly half of the plant and animal taxa showed a decreasing trend with increasing altitude while the other half showed hump-shaped or bimodal distribution patterns in Mt. Kilimanjaro [3]. However, most of the current studies have focused on plants and animals, ignoring the interaction between soil microorganisms and plants [4]. In addition, soils are believed to be exceptionally biodiverse parts of ecosystems [5]. As widespread mutualists, fungi are symbiotic with plant roots and affect the growth and distribution of plants [6,7], and the effect may be different in different environment gradients [8], which play an important ecological role in ecosystem functions.

As the most widespread mutualists, arbuscular mycorrhizal (AM) fungi can form symbionts with 80% of plant species [9–11], which play important ecological functions in maintaining ecosystem balance in all kinds of ecosystems [11,12]. Research showed

that AM fungi promote root growth and have positive effects on aboveground plant productivity through direct and indirect interactions [2]. Furthermore, AMF diversity was a key factor in maintaining plant biodiversity and ecosystem function [13–15]. Moreover, studying AMF biodiversity and distribution is the basis for predicting the evolution and succession of mountain ecosystems [16]. At present, more and more attention has been paid to the study of AMF diversity in mountain ecosystems. Yang et al. reported that elevations had a significant effect on AMF diversity and community distribution in the Qinghai-Tibet Plateau [17]. Gough et al. suggested that AMF are a ubiquitous group of soil microorganisms [18]. However, the measurement results of AMF diversity might be different using different methods in the same region. For example, Shi et al. (2014) researched AMF diversity and identified 63 AMF belonging to 12 genera by the traditional morphological identification method in Mt. Taibai [15], while Zhang et al. (2021) found 103 AMF species from soil samples, which belong to 19 genera using molecular identification method in Mt. Taibai [19]. It can be suggested that more AMF taxa are identified by the molecular method. Therefore, in this study, high-throughput sequencing molecular methods were used to explore AMF diversity and distribution in plant roots and investigate its influencing factors.

As the intersection of the flora of North China, Central China, and West China, Qinling Mountain is the natural dividing line between North and South China, with abundant species and resources. As the main peak of Qinling Mountain, Mt. Taibai is dominated by forest landscapes, rich in biological species, and is known as a green pearl in Western China [20]. The plant species is very rich in Mt. Taibai, which is one of the most abundant plant species in the temperate zone in China. Due to the different climatic gradients and the particularity of the vertical distribution of vegetation along altitudes, Mt. Taibai has become a natural place to study biodiversity [21].

Therefore, this study used molecular identification methods to explore the diversity and distribution mechanism of AMF associated with roots at different altitudes in Mt. Taibai, aiming to determine the biodiversity and variations of AMF with altitudes in the mountain ecosystem. It is expected to enrich the ecological theory of AMF by providing supporting data on different altitudes of mountain ecosystems.

## 2. Materials and Methods

### 2.1. Description of Study Region

This study was conducted in Mt. Taibai (23°49′31″–34°08′11″ N and 107°41′23″–107°51′40″ E) of the Qinling Mountain, which lies in the ecological transition zone between the subtropical zone and the warm temperate zone and is an important east-west mountain range across the central part in China. As the main peak of the Qinling Mountains, Mt. Taibai is the first peak in the east of the Qinghai-Tibet Plateau in China, with the highest altitude of 3771.2 m. The climate zone of Mt. Taibai is obvious, which is divided into a temperate monsoon climate zone (800–1500 m), a cold temperate monsoon climate zone (1500–3000 m), a subarctic climate zone (3000–3350 m), and frigid climate zone (>3350 m). Besides, Mt. Taibai is rich in plant species and has a special geographic location, complex and diverse climate, and large altitude gradient, as well as one of the most abundant temperate plant species in China. In addition, the distribution of vegetation in the vertical zone of Mt. Taibai is also very special, which is of great significance to the study of the distribution of vegetation in the north and south of China and provides an ideal environment to conduct scientific research [22]. The plant distribution from the bottom to the top of Mt. Taibai can be divided into deciduous broad-leaved forest belt, coniferous forest belt, alpine shrub belt, and meadow belt. The deciduous broad-leaved forest belt is mainly distributed by *Quercus variorum* forest, *Quercus aliena* var. *Acuteserrata* forest and *Betula albo-sinensis* forest. The coniferous forest belt is mainly distributed by *Abies fabri* forest and *Larix gmelinii* forest. The alpine shrub belt and meadow belt are above 3350 m, mainly distributed dwarf creeping shrub, dwarf meadow, mossy community, and lichen community (Table S1). Meanwhile, it has always been a hotspot for biodiversity research.

### 2.2. Collection of Samples

Twelve different altitudes were selected within the range of 663–3511 m in Mt. Taibai. At every target altitude, three 20 m × 20 m sample squares were set up, and the distance between each sample square was at least 50 m. In each sample square, a five-point sampling method was used to collect 0–30 cm soil, including all the plant roots and soil, and they were mixed as one sample. Three squares were considered three replicates. Finally, we separated all the mixed roots from the soil and put the mixed root samples and soil into different sealed bags, respectively.

Soil samples were used to determine the physical properties and nutrient elements after air-drying. All plant root samples were divided into the following three parts. The first part of each plant root sample was immediately stored in a −80 °C freezer for DNA extraction. The second part of each plant root sample was transported to the laboratory to carry out the determination of AMF colonization. The third part was stored in a 4 °C refrigerator to determine the nutrient elements, such as C, N, P, and C/N.

### 2.3. Bioinformatics Analysis of Sequence Data

Genomic DNA was extracted from plant root samples using the Fast DNA SPIN Kit for Soil (MP Biomedicals LLC, Santa Ana, CA, USA) according to the manufacturer's protocols. The extracted DNA was subjected to nested PCR by a thermocycler PCR system (GeneAmp 9700, ABI, Foster City, CA, USA). PCR amplification was performed with primers AML1F (5′-ATCAACTTTCGATGGTAGG ATAGA-3′) and AML2R (5′-GAACCCAAACACTTTGGTTTCC-3′) by an ABI GeneAmp® 9700 PCR thermocycler (ABI, CA, USA).

Purified barcoded amplicons were pooled in equimolar concentrations and paired-end sequenced on an Illumina MiSeq PE300 platform/NovaSeq PE250 platform (Illumina, San Diego, CA, USA) according to the standard protocols by Majorbio Bio-Pharm Technology Co., Ltd. (Shanghai, China). Microbial community sequencing was conducted by Shanghai Majorbio Bio-pharm Technology using the Illumina-MiSeq sequencing platform. The data were analyzed on a free online platform (Majorbio I-Sanger Cloud Platform, available online: http://www.i-sanger.com, accessed on 3 August 2021). Used Uparse (version 7.1) software platform to perform taxonomic analysis of OTU representative sequences at a 97% similar level.

### 2.4. Measurement of AM Colonization and Parameters of Soil and Plant Roots

The colonization of AMF was determined according to the method of Phillips et al. [23]. First, selected fresh fine roots were cleaned and wiped dry, then immersed in a test tube with a mass fraction of 10% KOH solution, and heated in a water bath at 90 °C for 30 min until the roots became relatively transparent. When the roots were relatively transparent, the lye on the roots was cleaned and the roots were soaked in a 5% CH3COOH for 5 min. They were dyed with 5% volume fraction acid acetic ink and heated in a water bath at 90 °C for about 30 min. As the roots were fully dyed by the acetic acid ink, they were cleaned and then put in lactic acid for color separation for approximately 30 min. Finally, the roots were cut at about 1 cm and placed on a glass slide (each glass slide has 15 roots). The glass slide was observed under a Motic BA310 microscope at 100–400 times magnification to survey the colonization status.

The concentration of total carbon and nitrogen were measured by an elemental analyzer (GC IsolinkFlash 2000; Thermo Scientific, Waltham, MA, USA) analyzer. The phosphorus content in plant roots is determined by the molybdenum-antimony colorimetric method [24].

### 2.5. Calculation of AM Colonization, Relative Abundance, and Occurrence Frequency

The percentage of root length colonized by AM fungal structures was estimated and calculated according to the grading criteria of Trouvelot et al. [25].

The relative abundance (species/OTU) of AM fungal genus was calculated as the percentage of the number of species/OTUs in each genus divided by the total number of species/OTUs in all genera, then take the average of the three samples.

The occurrence frequency of AM fungal genus was defined as the percentage of the number of samples where this genus was observed to the number of all samples in this genus.

Shannon–Weiner index: the $P_i$ of AMF species or OTUs was defined as the percentage of the sequences for each species or OTUs detected to total species or OTUs sequences in a sample;

$$H = -\sum [P_i \log_2 (P_i)]$$

The Sobs index of AMF species or OTUs was defined as the numbers of species or OTUs in a sample.

### 2.6. Statistics and Analysis of Data

The colonization ratio, colonization density, AMF diversity indies including Sobs and Shannon index, stoichiometric characteristics of plant roots, and edaphic factors were all statistically analyzed and curved estimated by SPSS 25.0. Then made the scatter charts through Origin 21.0. AMF relative abundance and occurrence frequency were analyzed based on genus level by Excel 2019. The heat map was presented to explore the relationships between the AMF community and environmental variables based on correlation analyses. And it is conducted on MajorBio Cloud's bioinformatics analysis cloud platform. Based on multiple linear regression analysis of the relationship between altitude, environmental factors, and AMF diversity.

## 3. Results

### 3.1. Arbuscular Mycorrhizal Colonization in Plant Roots at Different AltitudesSubsection

The colonization of AM varied from 0 to 100%, with an average of 55.03% from 663 m to 3511 m (Figure 1a). AM colonization showed a quadratic function trend with $R^2$ was 53.46% and *P* was less than 0.01. It suggested that altitude has a significant effect on AM colonization. The highest colonization of AM occurred at 1170 m and 1450 m. Meanwhile, the highest colonization density also occurred at 1170 m with 41.09% (Figure 1b). And the colonization density changed from 0 to 41.09%, with an average of 10.39% in Mt. Taibai. Besides, the colonization density formed the quadratic function with $R^2$ was 34.18%. Elevation had a significant effect on AM colonization density.

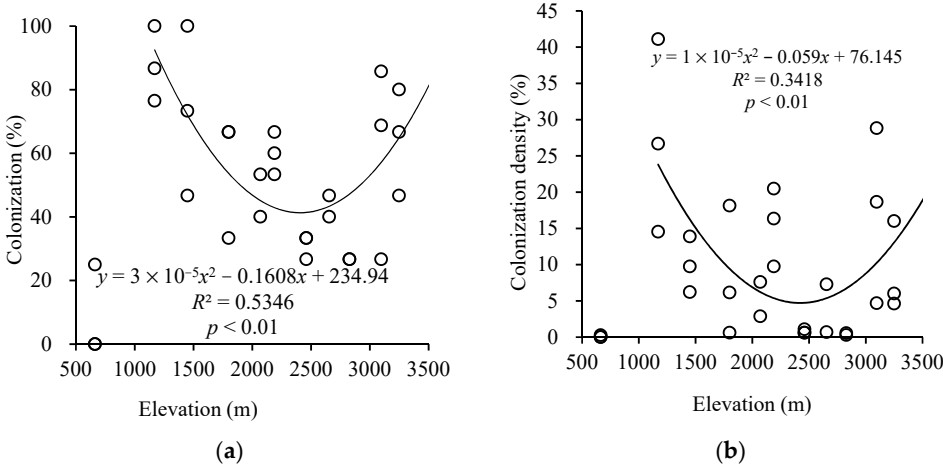

**Figure 1.** Change of arbuscular mycorrhizal colonization (**a**) and colonization density (**b**) in plant roots among different altitudes. Note: There are overlapping data points in (**a**,**b**).

### 3.2. AMF Community Composition and Distribution at Different Altitudes

A total of 287 OTUs belong to 62 species belonging to 8 identified and 1 unclassified genus representing four identified and one unidentified order (Table S2). Among them, *Glomus* was the dominant genus with the largest number of species and OTUs with 39 and 104, respectively. However, only 1 OTU was identified in the genus of *Pacispora*. In addition, the maximum number of species and OTU occurred at the altitude of 2190 m, with 38 and 166, respectively. While the minimum number of species occurred at 3097 m, and the minimum number of OTU was 25 and occurred at 2828 m.

The highest relative abundance of *Glomus* was 99.14%, which occurred at 1800 m (Figure 2). In addition, the relative abundance of *Glomus* exceeded 90% at the altitudes of 663 m, 1170 m, 1450 m, 1800 m, and 2070 m. At the higher altitudes of 2828 and 3250 m, the relative abundance of *Acaulospora* was higher with 49.14% and 56.52%. Besides, the genus of *Pacispora* only appeared above 2000 m and the relative abundance was the highest with 17.85% at the highest altitude of 3511 m.

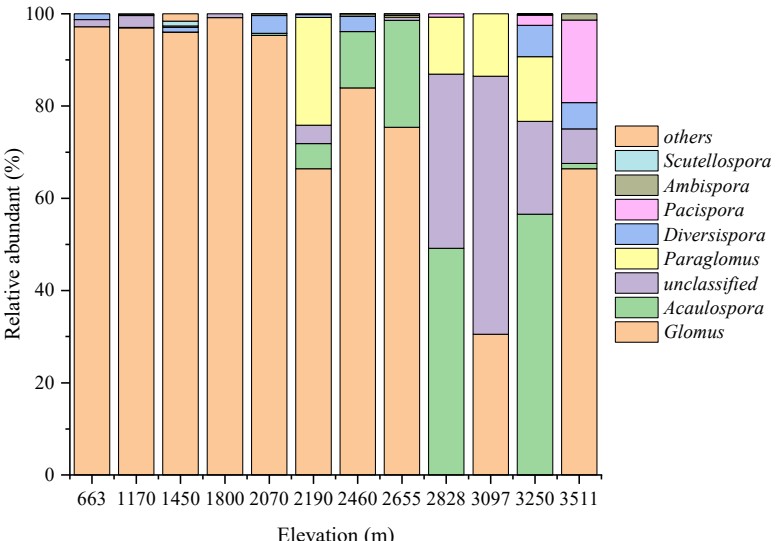

**Figure 2.** Relative abundance of genera identified based on different altitudes at the genus level. Note: The genus, whose abundance was less than 1%, were collectively classified as others.

### 3.3. Diversity of AMF in Mt. Taibai

AMF alpha diversity is expressed by the Sobs and Shannon indices (Figure 3). Both Sobs and Shannon indices showed the trends of increased first and then decreased with the increase of altitude based on the species and OTU level. Whether at species or OTU level, the highest Sobs and Shannon index both occurred at 2460 m. The highest Sobs index were 26 and 49 on the level of species and OTU, respectively (Figure 3a). At the same time, the highest Shannon indices were 2.68 and 2.08 at the level of species and OTU (Figure 3b). And the lowest alpha diversity indices both appeared at higher altitudes. With the increase in altitude, the changing trend of Sobs and Shannon indices could be simulated with quadratic function at the species and OTU level. What's more, elevation had a significant effect on alpha diversity indices.

AMF beta diversity was revealed by nonmetric multidimensional scaling analysis (NMDS) (Figure 4). The NMDS ordination resulted in a final stress value of 0.14 and 0.18 on species and OTU level, respectively. The results indicated that beta diversity also differed among the altitudes based on species and OTU level in Mt. Taibai.

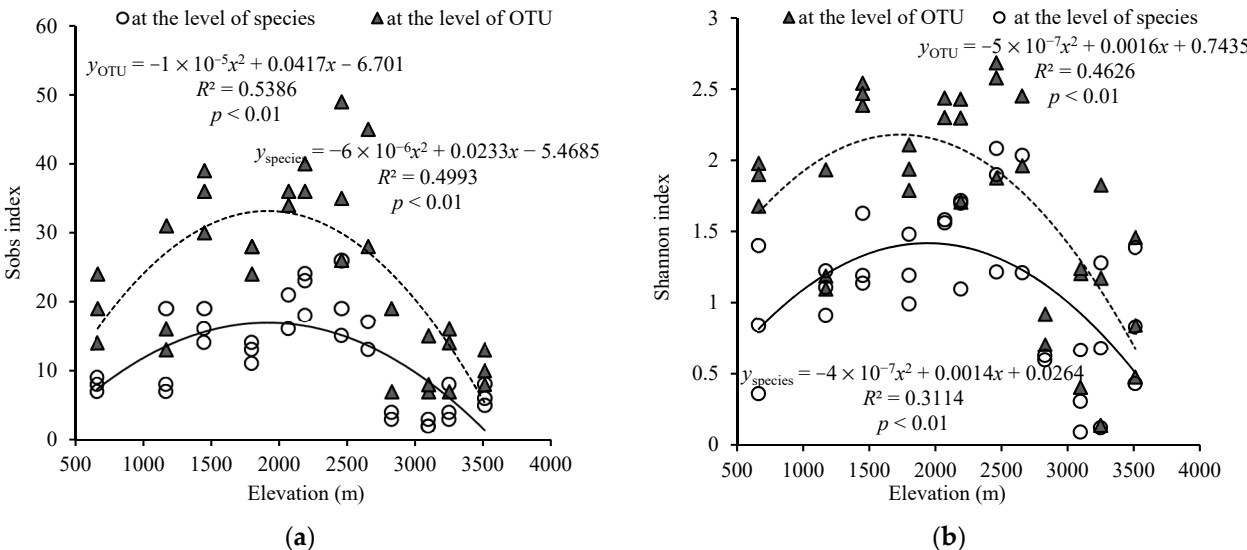

**Figure 3.** The variation of AMF is based on the species and OTU level among different elevations with the Sobs index of species (**a**) and the Shannon index of species (**b**).

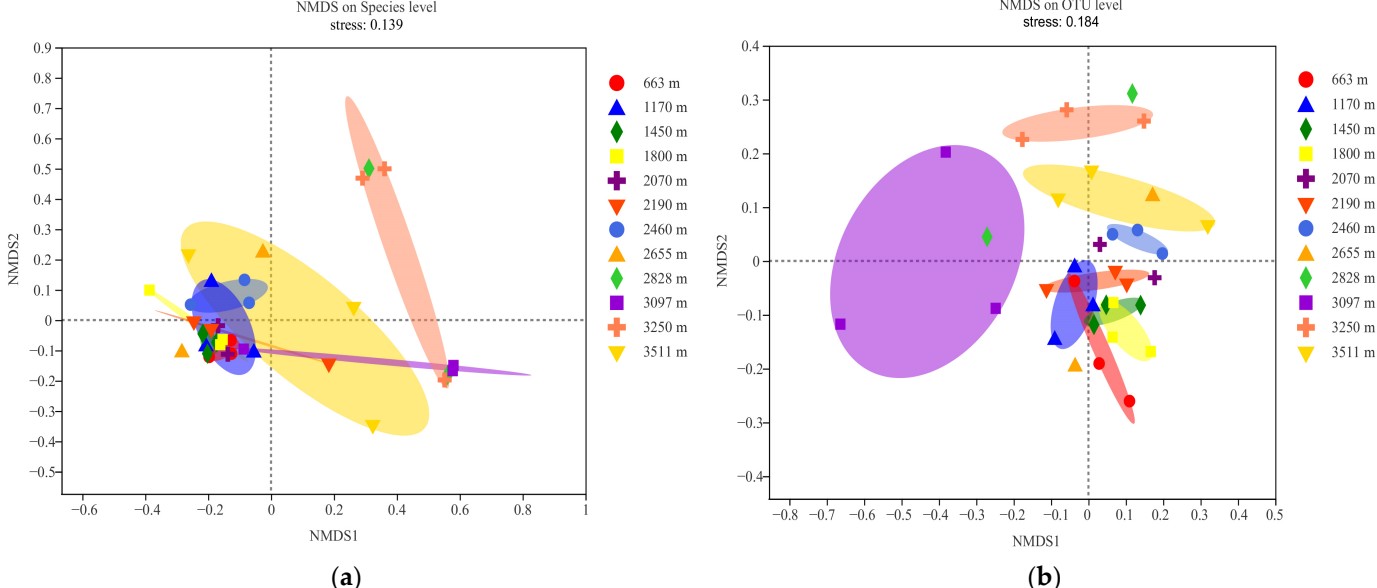

**Figure 4.** Nonmetric multidimensional scaling (NMDS) ordination of symbiosis AMF community composition of plant roots at different altitudes in Mt. Taibai ((**a**,**b**) stands for the species and OTU levels, respectively).

*3.4. The Relative Abundance and Occurrence Frequency of AMF Genus in Mt. Taibai*

It was found that the relative abundance ranged from 0.76% to 61.29% and from 0.24% to 53.58% based on species and OTU levels, respectively (Table 1). The fungi in the genus of *Glomus* were the most dominant, with the highest relative abundance of 61.29% and 53.58% based on species and OTU level, which was significantly higher than other genera. Meanwhile, the occurrence frequency of *Glomus* also was higher at 91.67%. *Unclassified in Glomeromycetes* was found in all altitudes with the highest occurrence frequency of 100%. The second abundant genus is *Acaulospora*, with a relative abundance of 9.46% and 5.32% at species and OTU levels, and the occurrence frequency was 66.67%. At the same time, the relative abundance of *Pacispora* was the same as *Ambispora* based on species level with 2.18%. In addition, the occurrence frequency of *Pacispora* was the same

as *Scutellospora* with 33.33%, and the occurrence frequency of *Diversispora*, *Acaulospora*, and *Paraglomus* were the same with 66.67%.

**Table 1.** Relative abundance and occurrence frequency of AMF genus in Mt. Taibai.

| Genus | Relative Abundance (Species)/% | Relative Abundance (OTU)/% | Occurrence Frequency/% |
|---|---|---|---|
| *Acaulospora* | 9.46b | 5.32 | 66.67 |
| *Ambispora* | 2.18 | 1.09 | 50.00 |
| *Archaeospora* | 5.72 | 1.82 | 41.67 |
| *Diversispora* | 4.36 | 3.58 | 66.67 |
| *Glomus* | 61.29 | 53.58 | 91.67 |
| *Pacispora* | 2.18 | 0.72 | 33.33 |
| *Paraglomus* | 4.96 | 5.45 | 66.67 |
| *Scutellospora* | 2.57 | 0.65 | 33.33 |
| *no rank* | 0.81 | 0.34 | 8.33 |
| unclassified in Diversisporaceae | 0.76 | 0.24 | 8.33 |
| unclassified in Archaeosporales | 2.18 | 0.85 | 16.67 |
| unclassified in Diversisporales | 1.36 | 0.37 | 16.67 |
| unclassified in Glomeromycetes | 2.18 | 25.97 | 100.00 |

*3.5. The Drive Factors of AMF Community and Diversity*

Among different genera, *Glomus* was most affected by soil and root nutrient factors (Figure 5). Both soil factors (pH and C/N) and root nutrients (C, N, and C/N) had a significant effect on *Glomus* through correlation analysis by heatmap at the genus level (Figure 5). *Unclassified in Diversisporaceae* and *unclassified in Archaeosporales* were both affected by root nutrient of N ($p < 0.01$) and ecological stoichiometry of C/N ($p < 0.01$). As for *unclassified in Glomeromycetes*, it was influenced by C/N ($p < 0.01$) and available phosphorus ($p < 0.05$) in soil. And soil and root nutrients had no significant effect on other AMF genera.

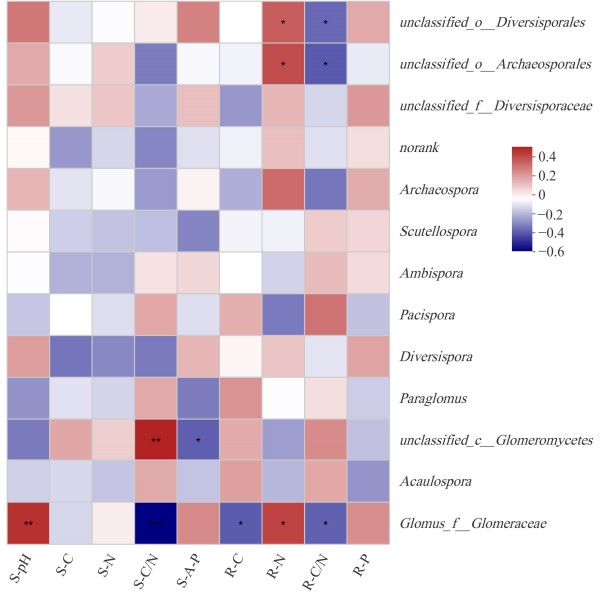

**Figure 5.** Influence of environmental factors on AMF genus. **Note:** *, **, *** indicate significant correlation at $p < 0.05$, $p < 0.01$, $p < 0.001$ confidence level, respectively.

Elevation had a positive and prominent effect on AMF Shannon diversity ($r = 0.493$) based on OTU level (Figure 6). Meanwhile, elevation had a positive effect on soil factors of pH ($r = 0.651$) and ecological stoichiometry of C/N ($r = 0.605$), plant factors, such as C ($r = 0.364$), N ($r = 0.379$) and C/N ($r = 0.345$). Besides, the AMF diversity index of Shannon

was greatly affected by soil and root nutrients, while the Sobs index was affected by soil and root ecological stoichiometry C/N and root N content.

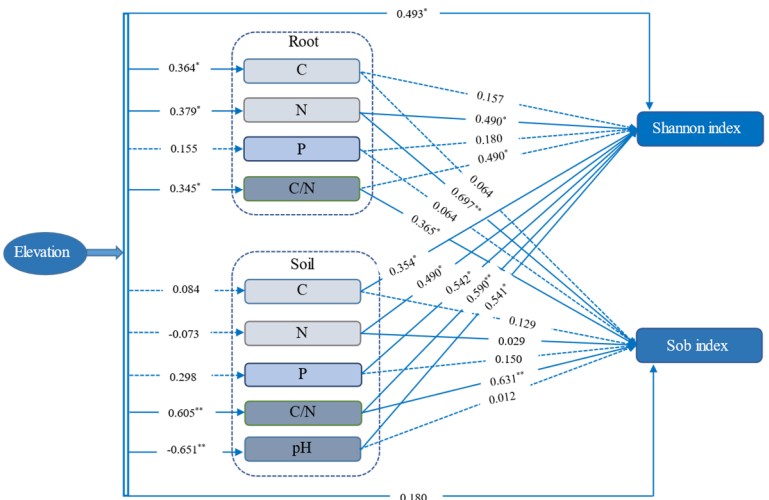

**Figure 6.** The relationships among elevation, soil factors (C, N, P, C/N, and pH), plant factors (C, N, P, and C/N), and AMF diversity indices are based on OTU level. Note: The effect of altitude on soil and root is expressed as a correlation coefficient, while the effects of soil and root factors on diversity indices are represented by standard regression coefficients. Blue solid represents significant positive or negative effects. Blue dashed represent nonsignificant paths. ** means $p < 0.01$; * means $p < 0.05$, respectively.

## 4. Discussion

The changes and laws of biodiversity along different environmental gradients are the important topic of biodiversity research [26,27]. Many environmental factors vary with altitude in mountain systems, so altitude is often used as an integrated factor to study plant and animal distribution patterns in mountain systems. In recent years, people have become more and more interested in knowing how microbes respond to the changes in environmental conditions because of their critical role in ecosystem functions [28–30]. Luo et al. also suggested that understanding the diversity of fungi in ecosystems may have predictive implications for biodiversity and ecosystem evolution processes [31]. Therefore, AMF diversity and community distribution along different altitudes were studied to explore the role of AMF in the mountain ecosystem and the responses to clime change.

In this study, the colonization rate and colonization density of AMF showed a trend of first increasing, then decreasing trends with the elevation. However, Gai et al. and Kotilinek et al. believed that AMF colonization showed a downward trend with increasing altitudes [32,33]. There are even studies that there were no significant differences in AMF colonization between high-altitude and low-altitude areas in the southeast of the Qinghai-Tibet Plateau [34]. The different results may be due to the differences in research sites or environmental factors, such as plant species, soil types, and so on. Liu showed that the arbuscular abundance of AMF was significantly influenced by altitude gradients [35], which was consistent with the results of our research. These different results also suggest that AMF could form a good symbiotic relationship with plant roots in the mountain ecosystem.

In the present study, the 287 OTUs and 62 species of AMF were identified and represented 8 identified genera, 5 unidentified genera in Mt. Taibai, which supported that AMF had a wide ecological range and was an important part of the ecosystem. Our research showed that 39 species belonged to the genus of *Glomus*, followed by *Acaulospora* with 5 species. Whether at the species or OTU levels, the relative abundance and the occurrence frequency of *Glomus* were the highest, which was consistent with the conclusions of most previous studies on the molecular diversity of AMF that *Glomus* was the dominant genus in the AMF community [19,36,37]. This may be due to its wide ecological range and the certain resistance in complex environments [38]. Besides, *Glomus* can usually produce large

numbers of spores and hypha fragments, which can extensively spread and colonize the roots of plants [37,39]. In terms of different altitudes, *Glomus* dominated at lower altitudes, whereas *Acaulospora* were more abundant at the higher altitudes of 3250–3511 m. This result was consistent with Oehl et al., who suggested that the genus of *Acaulospora* was more abundant in the highlands than in the lowlands in Switzerland [40]. Moreover, Haug et al. and Yang et al. also came to similar conclusions [8,41]. The different distribution at low and high altitudes in AMF explains the correlation of AMF species with altitude and suggests that there may be potential niche differentiation along the altitudinal gradient.

In addition to different distribution, the diversity of AMF was also different with the altitude change. It was discovered that AMF diversity indices of Sobs and Shannon showed the trends of quadratic function increasing first and then decreasing, whether on the level of species or OTU. However, Guo et al. and Egan et al. studied the AMF diversity and suggested that the alpha diversity decreased monotonically with the increase in altitude [21,42]. The reason for this phenomenon may be that this study was conducted in a large-scale altitude range of 663–3511 m, while Guo et al. and Egan et al. explored the AMF diversity in a relatively small altitude range. Therefore, in this study, the AMF diversity varies in different climatic environments. Moreover, some studies have proved that AMF diversity is closely related to plant richness [43], and plant richness is also different on different altitude gradients in Mt. Taibai (Table S1). This may also be the reason why AMF diversity shows different trends with increasing altitude in Mt. Taibai.

In this study, it was also found that the higher Shannon and Sobs indices appeared at mid-altitudes, whether based on species or OTU level. The highest diversity indices occurred at 2460 m, and altitude has a significant effect on them. Bonfim et al. showed that AMF diversity at higher altitudes was higher than at lower altitudes in the Atlantic forest system [44]. The reason for this phenomenon may be that the mid-altitudes have less human disturbance than the low altitude region and a less extreme climate environment than the high altitude region [45]. Therefore, the mid-elevation area is favorable for AMF sporulation and growth. Gai et al. and Shi et al. supported that altitude has no significant influence on AMF diversity [32,39]. Because altitude is a comprehensive factor, the effects of altitude on AMF diversity may be caused by differences in geographic location and environmental factors [15,46,47]. Besides, previous studies have shown that environmental factors, especially the geographical environment and soil factors, have an important impact on AMF diversity. Different ecological factors would affect the growth, development, colonization, and reproduction of AMF, which would cause differences in AMF diversity in different ecosystems [48–51]. Therefore, it is necessary to study the environmental factors of different altitudes further.

Determination of soil and plant nutrients found a significant impact on AMF diversity indices of Sob and Shannon. Our results were consistent with previous research conclusions that altitudes and soil variables had a significant impact on AMF diversity and richness [36]. Besides, Montiel-Rozas et al. [52] showed that AMF diversity and richness were only affected by soil properties, and soil factors were the main driving force for AM fungal communities. Our research found that whether it was in soil or plant roots, P concentration significantly affects the AMF Shannon index. This result was consistent with Maitra et al., who confirmed that the AMF Shannon diversity index showed a positive response to P [53]. Ceulemans et al. showed that AMF diversity decreased with the increase in soil P utilization [54]. Therefore, the change of P concentration is an important predictor of the response of AMF diversity to soil nutrients [55]. Previous studies have also shown that the addition of N increased AMF diversity in N-deficient soil [56–58], which was consistent with our study that N content of plant roots has a significant effect on AMF diversity, but N content in the soil had no effect on it. This result showed that plant roots had a greater impact on AMF diversity than soil. This also suggests that AMF tends to be symbiotic with plants to absorb nutrients, thereby increasing the diversity of mycorrhiza to increase the plant's own competitive advantage [59]. In addition, studies have suggested that the identity of the host plant has been considered to be one of the most important factors in

shaping AMF community composition [60–62]. It was speculated that the vital effects of the host plant on the AMF community might be related to the C/N in plant roots.

Moreover, it was found that the AMF community have different affinities with soil and root nutrient. And glomus is most affected by soil and root nutrient factors through the analysis of Heatmap. The genus of *unclassified in Diversisporales* and *unclassified in Archaeosporales* were significantly affected by root N concentration and C/N. However, the genus of unclassified *Glomeromycetes* were correlated with soil P concentration and C/N. The results revealed that there were different relationships between soil characteristics and the AMF genus, which is consistent with Kim et al. [63]. These also suggested that AMF taxa have different environmental preferences in tropical montane rainforests. Therefore, this also explained the different distribution of AMF communities at different altitudes. Besides, soil nutrient concentration also is an important factor in the AMF community. Zhao et al. [37] confirmed that soil nutrients have an impact on AMF communities, as a lack of nutrients inhibits spore germination and dissociation. Therefore, soil factors play an important role in AMF diversity and community in the mountain ecosystem.

## 5. Conclusions

In this study, the biodiversity and variation of AMF with plant roots at different altitudes were explored by molecular identification methods in Mt. Taibai. And it was found that there is abundant AMF diversity in the mountain ecosystem. Altitude has a significant impact on AMF diversity and community distribution. Whether it is species level or out level, Sob and Shannon indices show a quadratic equation changing trends with the increase of altitudes. In addition, whether in soil or plant roots, the ecological stoichiometry of C/N has a major impact on AMF diversity. Further, *Glomus*, as a dominant genus, was most affected by root and soil nutrients. These findings suggest that soil and root nutrient are important factors affecting AMF diversity and variation among different altitudes in the mountain forest.

**Supplementary Materials:** The following supporting information can be downloaded at: https://www.mdpi.com/article/10.3390/d14080626/s1, Table S1: Plant species at different altitudes in Mt. Taibai. Table S2: The distribution of AMF order, family, genus, and the number of AMF species aoutOTU among different altitudes.

**Author Contributions:** Conceptualization, M.Z., M.Y. and Z.S.; methodology, M.Z., M.Y. and Z.S.; software, M.Z.; validation, M.Z., M.Y., Z.S. and J.G.; formal analysis, M.Z.; investigation, M.Z., Z.S., J.G. and X.W.; resources, M.Z. and Z.S.; data curation, M.Z. and Z.S.; writing—original draft preparation, M.Z. and M.Y.; writing—review and editing, M.Z. and Z.S.; visualization, M.Z., Z.S. and X.W.; supervision, Z.S.; project administration, Z.S.; funding acquisition, Z.S. All authors have read and agreed to the published version of the manuscript.

**Funding:** This research was funded by NSFC (32171620, 31670499), Scientific and technological research projects in Henan province (192102110128), Program for Science & Technology Innovation Talents in Universities of Henan Province (18HASTIT013), Key Laboratory of Mountain Surface Processes and Ecological Regulation, CAS (20160618).

**Institutional Review Board Statement:** Not applicable.

**Informed Consent Statement:** Not applicable.

**Data Availability Statement:** The datasets presented in this study can be found in online repositories. The names of the repository/repositories and accession number(s) can be found below: SRA database and PRJNA843859.

**Conflicts of Interest:** The authors declare no conflict of interest.

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
