# Peer review of "Biodiversity and Variations of Arbuscular Mycorrhizal Fungi Associated with Roots along Elevations in Mt. Taibai of China"

_diversity, doi:10.3390/d14080626_

Round 1

Reviewer 1 Report

This paper presents original experimental data on arbuscular mycorrhizal fungi as related to the elevation along Mt Taibai in China..

However I recommend to refurbish/expand the § Materials & Methods.:

   - all parameters and formula need to be defined clearly in M&M.

   - clarity of figures ??????

    - some of the simulated equations as eg Fig. 1 are difficult to interpret, the more we are dealing here with second order statistics based on colonisation ratio and colonisation density

   - why do you not present a rough sketch of the altitudinal vegetation belts (see famous German naturalist Humboldt)

     - The proposed equation of "Arbuscular Mycorhizal Colonisation"  under § 2.5 in Materials and Methods is difficult to understand (mathematically). Do you want to discriminate colonisation surfaces or occurences or .....???

    - Fig 6 is a well intended synthesis chart which deserves high recognition. . However, it is unclear.

I would suggest to abide by the normal pathway analysis rules. I enclose a fertiling example on Teff from FAO fertiliser trials in Ethiopia I happened to analyze.

    Your difficulty is to choose (please consider it to be your hypothetical meaningful model) the main pathways, starting from either one (in your case you restricted it to only elevation) or several (perhaps you could add vegetation cover gradients, ...) starting points, ending eventually in terminal understandable parameters (be very cautious before using second order statistics). Why not use the simple "species richness" as one of your terminal points?

Make a clear difference between your partial standardized regression coefficients which will follow your hypothesized pathways, against simple correlation coefficients which will explain solely one of your factors along the pathways independently of the other choosen factors.

Author Response

Thank the reviewer's constructive suggestions. A point by point response to the reviewer's comments has been uploaded. Please see the attachment.

Reviewer 2 Report

Dear authors, I have read with interest the manuscript entitled "Altitude affects the diversity of arbuscular mycorrhizal fungi associated with roots mainly by affecting edaphic and root nutrient factors in mountain forests: A case study of Mt. Taibai in China". 

Your research is interesting for the field of AMF, which are very important symbionts for plants.

There are some suggestions that I consider will improve your work.

Title - I suggest a more condensed form of the title. You repeat the word "affect" two times. 

General comment - remove any personal words like "we", "our" etc, and make the text more impersonal.

The last paragraph of the introduction should present the clear aim of your work. You can present the three sentences between lines 77-83 as separate hypotheses.

The Material and Method section presents well the research methodology in separate sub-sections.

Both Results and Discussion sections are well written and clear, with a correct dimension of the text.

I suggest you to point better your findings in the conclusion section, with values. And each important finding in a separate sentence.

Overall, I like the manuscript and the way how the research was conducted.

Author Response

Thank the reviewer for her/his efforts and comments in our manuscript. A point-by-point response to the reviewer's comment has been uploaded. Please see the attachment.

Reviewer 3 Report

The study presented in this manuscript examines changes in alpha and beta diversity, as well as colonisation density, of arbuscular mycorrhizal fungi (AMF) along a substantial elevation gradient (~3000 m) at Mt. Taibai. The critical roles of mycorrhizal fungi on plant growth, community properties, and ecosystem functions, among many plant community characteristics, are increasingly being recognised. And, a wide range of studies have examined spatial variation in AMF in a variety of communities. Yet, comparatively few studies have examined AMF community structure along elevational gradients, despite the long history in community ecology of research on effects of elevation, and the central role these studies have played in the evolution of community ecology as a field. Coupled with the large elevation gradient studied, and the use of NGS to assess the AMF community, this study has the potential to be of broad interest among mycologists, community ecologists, forest managers, among others. In general, the data are compelling, but I have several concerns about the data analyses and interpretations, which I think currently limit the impact of this manuscript. These concerns, as well as a couple of minor points, are outlined in more detail below.

Title: To me, the title overstates the implications of this study. Because the study is observational, rather than experimental, as no variables were manipulated, spatial patterns in the data cannot be attributed to any specific variables. Rather, the data analyses can only suggest associations between variables, rather than demonstrate cause and effect. As a result, I disagree with the assertion that edaphic and root nutrient factors were the primary drivers of variation in diversity among sites.

Abstract: In general, the abstract seemed accurate and effective. As with the title, though, I felt the abstract overstated some of the results - specifically, that the structural model analysis demonstrated a cause and effect relationship between AMF diversity and soil and root nutrient factors (l. 27-28). As noted above, these results suggest an association, rather than demonstrate that these variables drove AMF diversity.

Introduction: This section seemed to provide a solid rationale for the study, including the results of previous studies of AMF community patterns on Mt. Taibai.

Materials and Methods: The descriptions of the methods seem to provide a good balance between brevity and detail. I do have several questions and comments, though, related to the statistical analyses.

First, if I understand correctly, four sets of samples were collected at each of 12 different elevations, leading to four replicates at each elevation (l. 102-106). However, the Methods also refers to three samples as replicates (l. 155), and Figure 1 shows only two to three replicates per elevation. (If there are overlapping data points in this figure, as, for example, at 600 m in Fig. 1b, this should be made clear in the legend.)

It’s not clear why both Sobs and Shannon were used as diversity indices. Also, the calculation used for Sobs should be made clear - this seems to be referring to S<sub>obs rather than the name of an index.

For the post-hoc tests, it’s not clear why Duncan’s multiple range was used. Particularly given the large number of comparisons involved, Duncan’s is a problem because it doesn’t control for the familywise error rate.

A fairly minor comment, but nearly all the % data are given to two decimal places, but the measurements don’t provide this level of resolution.

For Figure 1, there doesn’t seem to be any biological basis for why a cubic function provides the best fit, and I’m concerned that this may be a case of over-fitting a function to the data. In addition to possibility over-fitting the data, the cubic function appears to be driven by four data points, three of which are at ~ 1100 m, and one at ~ 1500 m.

The heat map shown in Figure 5 is presented as being based on correlation analyses (which seems like the correct approach). If so, then these patterns can only be presented as possible associations, and not as drivers of AMF variability. (See also my comment about the Title and Abstract, and my concern about assertions about cause and effect when the study was observational rather than experimental.) Also, which correlation analysis was used? Pearson’s, or something different?

Author Response

(The authors gave the same response as above.)

Reviewer 4 Report

The manuscript with the title “Altitude affects the diversity of arbuscular mycorrhizal fungi associated with roots mainly by affecting edaphic and root nutrient factors in mountain forests: A case study of Mt. Taibai in China”, presents diversity and distribution of AMF in plant roots at twelve different altitudes in Mt. Taibai (a plant diversity hotspot in China), using molecular identification methods. Results are important for understanding soil diversity associated with altitudinal gradients.

Abstract
There is no need to mark the start of the phrases with: (1) Background:, (2) Methods:, (3) Results:, (4) Conclusions:.

Line 62 “19 genus” replace with “19 genera”

Due to the close relationship between vegetation and AMF, I would like to suggest to the authors to include some paragraph at material and method or Results/Discussion section about the vegetation existing at those altitudes. The supplementary file 2 with the table of species at each altitude is highly relevant, but if it is not included in the body of the manuscript at least a short description of general characteristic plant communities/landscape could be part of the manuscript.  A map would also be highly suggestive.

Best regards.

Author Response

(The authors gave the same response as above.)

Round 2

Reviewer 3 Report

I want to start my response by thanking the authors for the considerable amount of time and thought that went into addressing the reviewer comments. It appears the authors have addressed all my major concerns, and I don't have any additional major concerns.